# Personalization Toolkit: Training Free Personalization of Large Vision Language Models

**Soroush Seifi**\*   **Vaggelis Dorovatas**\*   **Matteo Cassinelli**\*   **Fabien Despinoy**   **Daniel Olmeda Reino**   **Rahaf Aljundi**

*Toyota Motor Europe*

**Reviewed on OpenReview:** *https://openreview.net/forum?id=5mbn3BOO29*

## Abstract

Personalization of Large Vision-Language Models (LVLMs) involves customizing models to recognize specific users or object instances and to generate contextually tailored responses. Existing approaches rely on time-consuming training for each item, making them impractical for real-world deployment, as reflected in current personalization benchmarks limited to object-centric single-concept evaluations. In this paper, we present a novel training-free approach to LVLM personalization called PeKit. We introduce a comprehensive, real-world benchmark designed to rigorously evaluate various aspects of the personalization task. PeKit leverages pre-trained vision foundation models to extract distinctive features, applies retrieval-augmented generation (RAG) techniques to identify instances within visual inputs, and employs visual prompting strategies to guide model outputs. Our model-agnostic vision toolkit enables efficient and flexible multi-concept personalization across both images and videos, without any additional training. We achieve state-of-the-art results, surpassing existing training-based methods.

## 1 Introduction

Large Vision Language Models (LVLMs) Liu et al. (2023a;b); Chen et al. (2024); Zhu et al. (2023); Li et al. (2023c); Agrawal et al. (2024); Wang et al. (2024) have demonstrated impressive capabilities in reasoning about visual content and answering visual questions across various domains. This suggests a great potential for deployment as visual assistants that can support users in their daily lives. However, current LVLMs are designed to provide generic, user-independent responses and recognize objects at the category level (Fig. 1, left).

The task of personalizing vision-language models was introduced by Alaluf et al. (2024) to enable LVLMs to recognize specific object instances and answer relevant questions accordingly. Existing approaches Alaluf et al. (2024); Nguyen et al. (2024) rely on training for a specific personalized object, diverting the LVLM from its original capabilities and incurring a large computational cost. Follow-up approaches attempted at replacing test-time training with large scale pretraining for the personalization task Pham et al. (2024); Pi et al. (2024), however, neither their effectiveness nor their scalability to various backbone models has been clearly demonstrated.

In this work, we argue that LVLM personalization can be approached without retraining the model parameters. We introduce a training-free approach that builds upon the strengths of pre-trained vision foundation models, the emerging capabilities of LLMs with in-context learning Dong et al. (2024) and retrieval-augmented generation (RAG) Lewis et al. (2020). Our method localizes instances using open-world object detectors Liu et al. (2023c); Kirillov et al. (2023); Oquab et al. (2023) and stores reference instance-level

---

\*Providing contracted services at Toyota Motor Europe.

Figure 1: Illustration of the personalization task and PeKit. Left: Without personalization, VLMs often fail to resolve named object references, leading to ambiguous responses. Right: PeKit personalizes VLMs by (1) extracting patch-level features from a reference image into a RAG memory, (2) matching them with object proposals in the query image to retrieve the target object, and (3) guiding the VLM via visual prompting using the detected object's bounding box and name. PeKit is VLM-agnostic, supports multi-concept personalization, handles video input, and achieves state-of-the-art (SOTA) performance, enabling capabilities often unsupported by prior methods.

features in memory banks, alongside their name and context. During inference, our retrieval module queries the memory bank and visual-prompts the LVLM. An overview of our approach is shown in Fig. 1.

With regards to the evaluation of personalization methodologies, existing benchmarks primarily focus on object-centric, single-concept tasks with the personalized instance prominently in the image, falling short of capturing the complexity of real-world applications where an AI assistant needs to understand multiple users and their belongings in dynamic scenes and environments. To address this gap, we introduce a novel and challenging benchmark derived from a video personalization dataset Yeh et al. (2023). Our benchmark not only features challenging single concept tasks but complex multi-concept interactions in addition to video question answering. We employ this benchmark to rigorously evaluate our method showing its effectiveness across diverse scenarios.

The key contributions of our work are: 1) We show that LVLM personalization is possible without training, enabling fast deployment. 2) We present a flexible method that supports multi-concept and video personalization across LVLMs using vision foundation models, RAG, and visual prompting. 3) We introduce a challenging benchmark that exposes current limitations and guides future research.[1] 4) We achieve state-of-the-art performance on various tasks across both existing datasets and our proposed benchmark, consistently outperforming previous methods.

We discuss the related work in Section 2 followed by an introduction to our vision toolkit for LVLMs personalization in Section 3. We then present our real-world benchmark and evaluate our approach in Section 4, and conclude in Section 6.

---

[1]Our benchmark and code will be available on Github.

## 2 Related Work

**Text-to-Image Personalization.** Personalizing text-to-image generation—i.e., generating images of a specific entity in novel contexts given a reference view—has been extensively explored. Early methods such as Textual Inversion Gal et al. (2022), DreamBooth Ruiz et al. (2023), and HyperDreamBooth Ruiz et al. (2024) achieve personalization by fine-tuning diffusion models for each entity, which limits scalability. More recent approaches, including InstantBooth Shi et al. (2024), JeDi Zeng et al. (2024), and Imagine He et al. (2024), circumvent this issue by pretraining for personalization, thereby eliminating the need for test-time fine-tuning.

Beyond generation, several works have explored personalized image retrieval using CLIP Radford et al. (2021) as the backbone. For example, PALAVRA Cohen et al. (2022) learns new concept tokens from a few user-provided images while keeping CLIP frozen, SEARLE Baldrati et al. (2023) maps reference images to pseudo-word tokens via a lightweight network and combines them with textual descriptions of desired changes for the Zero-Shot Composed Image Retrieval (ZS-CIR) task, and ConCon-Chi Rosasco et al. (2024) fuses image and text features through a learned composition network. In all these cases, retrieval is performed based on text–image similarity in CLIP space.

While these approaches focus on either text-to-image generation or retrieval, our work addresses a related but distinct challenge: identifying the same personalized object across different images and enabling personalized interactions with those objects through Large Vision–Language Models (LVLMs) such as LLaVA Liu et al. (2023b). We further posit that, unlike text-to-image personalization approaches that typically require training, personalization within LVLMs can be achieved without any adaptation of the underlying large language model.

**Large Vision–Language Model Personalization.** Personalizing LVLMs was first introduced in MyVLM Alaluf et al. (2024), which trains a concept head for specific objects on top of the CLIP `CLS` token. Similar to DreamBooth Ruiz et al. (2023), MyVLM uses rare tokens to encode personalized concepts, but this can introduce unintended behavior in language assistants and requires optimizing the LLM captioning loss for personalized conversations. Yo'LLaVA Nguyen et al. (2024) improves upon MyVLM by adding a dedicated token to the LLM head for each personalized object, learning concept tokens to describe them. However, this creates a challenging incremental classification problem De Lange et al. (2021), and like MyVLM, it requires test-time training for each new concept, limiting scalability to one concept at a time.

To avoid test-time training, recent works leverage large-scale pretraining. PVIT Pi et al. (2024) fine-tunes on synthetic personalized dialogues and uses reference images during inference. PLVLM Pham et al. (2024) aligns CLIP `CLS` and DINOv2 Oquab et al. (2023) embeddings with the LLM. Both works, primarily target individuals and do not support object-level personalization. Besides, personalization remains query-specific and is largely limited to VQA tasks.

In contrast, our approach introduces a modular, training-free framework that requires no retraining, scales naturally to multi-concept and video scenarios, and avoids relying on pre-trained tokens or reference images at inference time.

Finally, a concurrent work—Training-Free Personalization via Retrieval and Reasoning on Fingerprints Das et al. (2025)—also proposes a training-free framework. It derives textual and visual fingerprint attributes from reference images and employs multi-step reasoning to recognize personal concepts. While effective, this method depends on a complex, hand-engineered pipeline involving attribute extraction, chain-of-thought reasoning, cross-modal verification, and pairwise comparisons, making it vulnerable to hallucinated or noisy textual attributes generated by the underlying LVLM. By contrast, our approach uses a simpler patch-level matching strategy that avoids reliance on potentially hallucination-prone textual descriptions.

**Visual Prompting.** It represents the usage of visual cues such as bounding boxes or arrows to guide Vision-Language Models. CLIP Radford et al. (2021) interprets these marks to modify its `CLS` token embedding accordingly Shtedritski et al. (2023). Set of Mark Prompting Yang et al. (2023) integrates GPT-4V with visual prompts using tools like MaskDINO Li et al. (2023b), SAM Kirillov et al. (2023), Semantic SAM Li et al. (2023a), and SEEM Zou et al. (2024). ViPLLaVA Cai et al. (2024) enhances LLaVA Liu et al. (2023b)

to follow visual prompts by tuning on GPT-4V-labeled data. Contrastive Region Grounding (CRG) Wan et al. (2024) improves LLaVA's focus on objects by contrasting token probabilities with and without target object masking. Our experiments show that LLaVA and other LVLMs can describe objects accurately with proper instruction and context. Training-free methods like CRG Wan et al. (2024) can further enhance attention if needed.

# 3 Approach

This section outlines our personalization toolkit, coined as *PeKit*, for enabling any LVLM to perform personalized detection and answer generation. We employ a three-stage pipeline: **View Extraction** to extract robust object-level features from reference images and store them in a memory module, **Personalized Objects Retrieval** to identify objects in the query image, and **Personalized Answer Generation** using visual prompting to generate user tailored and contextualized responses. We refer to Fig. 1 for an illustration of our approach.

## 3.1 Preliminary

We consider a given LVLM, a large language model with visual understanding capabilities and a set $P$ of all personalized objects introduced to the LVLM. Each object $p \in P$ is associated with a set of reference images $\{I_p\}$, a name or identifier $n_p$ and optionally $c_p$ a context of the object. Our objective is to generate a personalized response for all images containing $p$ during inference, while producing a general caption for any other image that does not contain any of the personalized objects. The LVLM, e.g., LLaVA Liu et al. (2023b) typically takes as input an input image $I_p$, a text query $Q$ and additional text as context or instruction.

## 3.2 Training-free View Extraction

Existing LVLM personalization techniques depend on image-level representations of the objects' training views Alaluf et al. (2024); Nguyen et al. (2024), which can lead to overfitting to the background of each object in the reference images, particularly for training-based approaches. To avoid such a bias, our method first localizes the object in the image and extracts only its corresponding features. We utilize an open-vocabulary segmentation network $F_{\text{ext}}$ to extract object-level masks $S_p$ based on each object's generic category $k_p$ which can be deduced from the name or the context[2]

$$S_p = F_{\text{ext}}(I_p, k_p). \tag{1}$$

We construct the average embedding vector $\mathbf{e}_p$ for object $p$ by average pooling of the embedding vectors produced by the image encoder $F_{\text{emb}}$ on the image $I_p$ over the region defined by the object-level mask $S_p$:

$$\mathbf{e}_p = \text{AvgPool}(F_{\text{emb}}(I_p), S_p) \in \mathbb{R}^{D_h}. \tag{2}$$

Considering $N$ reference images, we concatenate all object embedding vectors $\mathbf{e}_p^i$ (of the object $p$ pooled over the $i$-th reference image $I_p^i$) into a matrix $E_p = \left[\mathbf{e}_p^1, \ldots, \mathbf{e}_p^N\right] \in \mathbb{R}^{D_h \times N}$.

**Memory module.** After extracting the personalized objects' embeddings, we store each object's relevant properties in our memory module. The memory module is represented by a set $\mathcal{M}$ of object-specific entries:

$$\mathcal{M} = \{(E_p, (n_p, c_p))\}_{p \in P}, \tag{3}$$

where $n_p$ is the identifier or the name of the personalized object $p$, and $c_p$ is the context of the object, which can contain prior knowledge such as characteristics, background story or even relation to other personalized objects. When the number of personalized objects scales, the memory module $\mathcal{M}$ is easily converted into a Vector database, where nearest neighbor approximate search is deployed to retrieve instances matching a given query Han et al. (2023) ensuring efficiency and scalability.

---

[2]Although using the semantic category $k_p$ is recommended for optimal performance, our method attains state-of-the-art results even without it, relying solely on the generic category 'main'. See section 4.7.1 for further discussion.

### 3.3 Personalized Object Retrieval

During inference, our goal is to determine whether a personalized object is present in the provided image $I$. We use any available object proposal technique $F_{\text{prop}}$ to generate a set of proposals (e.g., bounding boxes) $O = \{o_i\} = F_{\text{prop}}(I)$ for potential object occurrences within the image $I$. Then for each proposal $o_i$, we calculate its object-level average embedding vector:

$$\mathbf{e}_{o_i} = \text{AvgPool}(F_{\text{emb}}(I), o_i). \tag{4}$$

We define the retrieval module $\mathcal{R}$ that takes an object embedding vector $\mathbf{e}_{o_i}$ and retrieves the memory entry $(E_j, (n_j, c_j))$ for a matching object $j$ as:

$$\mathcal{R}(\mathbf{e}_{o_i}) = \begin{cases} (E_j, (n_j, c_j)), \\ \quad \text{if } j = \arg\max_l \left\{ \text{sim}(E_l, \mathbf{e}_{o_i}) \right\} \\ \quad \text{and } \text{sim}(E_j, \mathbf{e}_{o_i}) > \tau \\ \phi, \text{ otherwise} \end{cases} \tag{5}$$

where $\text{sim}(E_l, \mathbf{e}_{o_i}) = \arg\max_k \left\{ \text{sim}(\mathbf{e}_l^k, \mathbf{e}_{o_i}) \right\}$. We calculate the proposal's similarity to the embeddings of the training views for all personalized objects. Any similarity measure (e.g., cosine similarity) can be employed for this purpose. We set a constant threshold $\tau$ to identify the personalized objects. We discard the object proposals in which no matching object is found by the retrieval module $\mathcal{R}$. Our method inherently supports the detection of multiple personalized objects.

### 3.4 Personalized Answer Generation

Once a personalized object is identified, our method generates captions specifically about that object, distinct from general captions a standard LVLM would produce. This involves emphasizing the detected object and incorporating prior knowledge about it. We achieve this through visual prompting by overlaying bounding boxes on the image and querying the LVLM to generate captions or answer questions focused on these objects. We use distinctive colors to differentiate recognized objects. We provide the LVLM with the object identifier $n_j$ (e.g., the instance name) and possibly a context $c_j$ for each personalized object. The LVLM incorporates this context and responds to queries using the given name $n_j$. For multiple personalized objects, we instruct the LVLM for each bounding box, name, and context. We refer to Appendix C.1 for the exact prompt format.

### 3.5 Video-personalization

To conduct personalized video question answering (VQA), we apply our method to the video frames. We first identify any presence of a personalized object, then each personalized object is consistently visually prompted using the same bounding box color across all frames to maintain identity. The video frames—including those with visual prompts and those without any detected personalized objects—are fed into a video large language model.

The model is then prompted to generate captions or answer questions using each detected object's identifier $n_j$, an optional context $c_j$, and the unique bounding box color used for the visual prompting of that object instance (Appendix C.1).

### 3.6 Choice of Vision Tools

We deploy GroundedSAM Ren et al. (2024) as the segmentation network $F_{\text{ext}}$ and ablate using GroundingDINO Liu et al. (2023c), where the mask is represented by the object's bounding box. We use DINOv2 Oquab et al. (2023) as the image encoder $F_{\text{emb}}$ to extract patch-level features of the objects and ablate using CLIP Radford et al. (2021) in Appendix A.1. During inference, GroundingDINO Liu et al. (2023c) is queried with the term 'object' to generate object proposals.

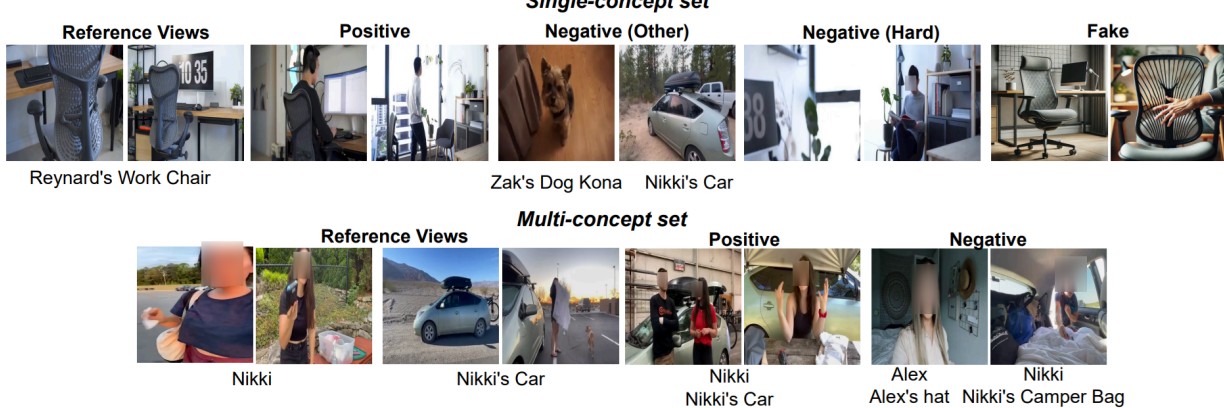

Figure 2: Our proposed evaluation set This-Is-My-Img, built on the This-Is-My dataset Yeh et al. (2023): Example reference views and validation samples from the single-concept category 'Reynard's Work Chair' and the multi-concept category 'Nikki' and 'Nikki's Car'.

## 4 Experiments

In this section we present the considered benchmarks and introduce our LVLM personalization benchmark. We then compare our approach with SOTA personalization methods and baselines under various settings.

### 4.1 Existing Benchmarks

We consider the datasets from **Yo'LLaVA** Nguyen et al. (2024) and **MyVLM** Alaluf et al. (2024). Yo'LLaVA includes 40 categories of objects, buildings, and people, with 4–10 images per category for training and validation. It also features a VQA benchmark with multiple-choice questions (A/B). MyVLM comprises 29 object categories, each with 7–17 images, using 4 images for training and the rest for validation, with final accuracies averaged over 5 runs. For both datasets, we use only reference images and the semantic category of each object for training view extraction (see 3.2), discarding all other data, such as the ground-truth captions and negative images.

### 4.2 Real-world Personalization Benchmark

**Yo'LLaVA** Nguyen et al. (2024) and **MyVLM** Alaluf et al. (2024) focus mainly on object-centric scenes under controlled conditions, lacking real-world complexity such as background clutter, occlusion, and diverse contexts—limiting their suitability for evaluating robust LVLM personalization.

We introduce the **This-Is-My-Img** benchmark, derived from the *This-Is-My* dataset Yeh et al. (2023), originally designed for video-level detection of personalized objects in realistic environments. Our benchmark includes several splits targeting specific personalization capabilities.

**Reference Views:** Five reference frames per object extracted from training segments. Unlike prior datasets, objects here may be partially visible, distant, or poorly lit (see Fig. 5, Appendix).

**Single-Concept Eval Set:** Set of images designed to evaluate a single personalized object. Fig. 2 shows examples. For each personalized object, images are labeled as:

- *Positive*: Includes only images where the personalized object appears and is used to test how well the method can correctly detect all occurrences of that object, that is, recall or *positive accuracy* in **Yo'LLaVA** Nguyen et al. (2024) and **MyVLM** Alaluf et al. (2024).

- *Negative (Hard)*: Images taken from video segments of the personalized object **where the object is not visible**. This tests method's overfitting to the background associated with the personalized object in the reference views.

- *Negative (Other)*: Images sourced from video segments of other personalized objects. This set measures method's robustness to intra-instance and the general dataset bias.

- *Negative (Fake)*: AI-generated images (via GPT-4o) that mimic the personalized object in different perspectives and environments while making sure the specific characteristics of the fake object are different from the real personalized object. These samples are used to assess the robustness of the methods against high visual similarity combined with a distribution shift.

- *Single-Concept VQA:* Multiple choice questions based on positive images, following the Yo'LLaVA benchmark VQA format.

**Multi-Concept Set:** This subset comprises images containing pairs of personalized concepts, such as person–object pairs or person–person pairs, fig 2. We extend the personalized instances from the original This-Is-My dataset with additional instances that frequently co-occur with them. For each pair, we collect all images containing both personalized concepts and partition them into two subsets. Positive samples include images in which both concepts appear simultaneously, whereas negative samples comprise images from all other pairs. Illustrative examples are presented in Fig. 2.

- *Multi-Concept VQA:* This component provides open-ended questions associated with each image in the multi-concept set, addressing both personalized concepts depicted. The set includes 55 images and their corresponding QA pairs.

**Video-QA Set:** Original video clips from This-Is-My dataset are coupled with open-ended questions regarding the personalized object in each video, designed to evaluate both temporal reasoning and the method adaptability to video-based inputs.

Overall, This-Is-My-Img offers a comprehensive and realistic benchmark for LVLM personalization, requiring accurate recognition and reasoning over personalized objects in cluttered and dynamic conditions. See Appendix B for more details on our proposed benchmark.

### 4.3 Metrics and Evaluation Setting

**Single-Concept Metrics.** We adopt the metrics and notation from Yo'LLaVA Nguyen et al. (2024) and MyVLM Alaluf et al. (2024). We report *Positive Accuracy (Recall)*—the proportion of correctly identified positives, *Negative Accuracy (Specificity)*—correctly identified negatives, and *Weighted Accuracy*, their average across $n$ personalized objects. We also include *Precision*, the ratio of true to predicted positives, averaged over $n$ objects, to capture the accuracy–precision trade-off. For single-concept VQA, accuracy is the percentage of correctly answered multiple-choice questions. Following MyVLM, we additionally report *CLIPScore* (caption–image similarity) and *Personalization Recall*, the fraction of captions correctly mentioning the target concept.

**Multi-Concept Metrics.** Positive accuracy measures images where both personalized objects are correctly detected; negative accuracy covers those where at least one is correctly **not** detected.

**Video and Multi-Concept VQA Metric.** For open-ended VQA, we use an autograding protocol Maaz et al. (2023) with ChatGPT-3.5 to assess contextual similarity between predicted and ground-truth answers (see Appendix C.2).

**Implementation Details.** The modularity of our method makes it generic regarding the choice of the LVLM model. We use LLaVA1.5-13B Liu et al. (2023a) as our primary LVLM model for consistency with previous works. We pair PeKit with InternVL2-26B Chen et al. (2024) , a state-of-the-art LVLM to ablate. We use LLaVA-OneVision-Qwen2-7B as our default video model and ablate with InternVideoChat2.5 as a superior video model. We utilize Cosine Similarity with a constant threshold of $\tau{=}0.75$ for detecting

Table 1: Visual recognition performance (%) on existing datasets. PeKit achieves state of the art performance without training. Cited entries are taken from their respective papers.

| **MyVLM Dataset** | | | | |
| --- | --- | --- | --- | --- |
| Method/Metric | Precision | | Accuracy | |
| | | Positive | Negative | Weighted |
| MyVLM Alaluf et al. (2024) | – | 96.6 | 90.9 | 93.8 |
| Yo'LLaVA Nguyen et al. (2024) | – | 97.0 | 95.7 | 96.4 |
| PeKit (G-DINO) | 79.1 | 94.3 | 98.8 | 96.5 |
| PeKit (G-SAM) | **82.3** | **97.6** | **99.0** | **98.3** |
| **Yo' LLaVA Dataset** | | | | |
| Method/Metric | Precision | | Accuracy | |
| | | Positive | Negative | Weighted |
| Yo'LLaVA Nguyen et al. (2024) | – | **94.9** | 89.8 | 92.4 |
| PeKit (G-DINO) | **77** | 89.9 | **98.9** | 94.4 |
| PeKit (G-SAM) | 74.8 | 91.0 | 98.7 | **94.9** |

Table 2: Visual recognition accuracy (%) on the This-Is-My-Img dataset. MyVLM and Yo'LLaVA produce many false positives in a challenging real-world scenario.

(a) Single-concept results.

| Method/Metric | Precision | | Accuracy | | | Avg. |
| --- | --- | --- | --- | --- | --- | --- |
| | | Pos. | Other | Hard | Fake | |
| MyVLM | 8.0 | **88.1** | 14.6 | 4.7 | 54.2 | 33.9 |
| Yo'LLaVA | 42.1 | 87.1 | 84.4 | 61.9 | **61.4** | 67.3 |
| Ours | **90.1** | 69.0 | **99.9** | **96.0** | 59.3 | **82.8** |

(b) Multi-concept results.

| Method/Metric | Precision | Accuracy | | Avg. |
| --- | --- | --- | --- | --- |
| | | Pos. | Neg. | |
| Yo'LLaVA | 10.0 | **83.6** | 25.0 | 54.3 |
| Ours | **96.1** | 45.4 | **99.8** | **72.6** |

personalized objects across all datasets (3.3). Refer to Appendix A.6 for further details on compute and memory overhead of our personalization toolkit.

## 4.4 Visual-Recognition

**Previous Datasets** Tab. 1 compares our method to MyVLM Alaluf et al. (2024) and Yo'LLaVA Nguyen et al. (2024) on their respective benchmarks. On MyVLM benchmark, PeKit achieves a new SOTA, improving both positive and negative accuracy with an average gain of 1.9%. On Yo'LLaVA benchmark, it increases negative accuracy by 8.9%, with slightly lower positive accuracy, resulting in a 2.5% improvement in weighted accuracy. While PeKit may miss some difficult instances, its low false positive rate helps avoid incorrect personalized outputs, favoring generic captions when uncertain. We ablate the choice of an open-world object detector, G-DINO Liu et al. (2023c), compared to the open-world semantic segmentation model, G-SAM Ren et al. (2024), for $F_{\text{ext}}$ in Eq. 1. The results show that our method outperforms previous methods in both cases. However, the more precise segmentation model, which extracts only patches of the object of interest, achieves the best accuracy on average.

Table 3: Captioning metrics on MyVLM dataset. Higher is better.

| Method | CLIPScore | Personal. Recall |
|---|---|---|
| MyVLM Alaluf et al. (2024) | 27.6 | 94.76 |
| PeKit (LLaVA) | **30.2** | **97.1** |

Table 4: Single-concept VQA accuracy on Yo'LLaVA and This-Is-My-Img datasets.

| Yo'LLaVA VQA Accuracy | | This-Is-My-Img Single-Concept VQA Accuracy | |
|---|---|---|---|
| LLaVA Nguyen et al. (2024) | 89.9 | LLaVA | 72.8 |
| Yo'LLaVA Nguyen et al. (2024) | 92.9 | Yo'LLaVA | 67.1 |
| PeKit (LLaVA) | 93.4 | PeKit (LLaVA) | 77.1 |
| PeKit (InternVL) | **95.9** | PeKit (InternVL) | **84.2** |

**This-Is-My-Img Benchmark.** Tab. 2 summarizes benchmark performance. For single-concept images, **MyVLM** achieves high positive accuracy but poor negative accuracy (14.6% on *other*, 4.7% on *hard*), indicating a bias toward positive responses and reliance on scene context over object details. **Yo'LLaVA** performs similarly but better (87.1% positive, 84.8% on *other*, 61.9% on *hard*). In contrast, **PeKit** attains much stronger negative accuracy (99.9%, 96.0%) and 90.1% precision—48% higher than Yo'LLaVA —showing far fewer false positives.

On fake images, differences narrow. All models struggle with stylistically similar objects; image-level approaches like MyVLM and Yo'LLaVA slightly better detect domain shifts. Nonetheless, PeKit remains robust to domain variation despite some difficulty with near-identical objects, achieving the best overall balance with 82.86% average accuracy, surpassing Yo'LLaVA (67.38%) and MyVLM (33.92%).

In multi-concept settings, PeKit shows slightly lower dual-object accuracy but far higher precision and negative accuracy, yielding an 18.3% overall gain over Yo'LLaVA. These results highlight our training-free method's robustness and low false-positive rate, especially in complex, real-world scenarios.

Overall, this evaluation underscores the challenge of personalization beyond object-centric imagery, advancing toward realistic benchmarks for intelligent visual assistants.

### 4.5 Visual-Question Answering

Tabs. 3, 4, and 5 evaluate PeKit on personalized object VQA across multiple scenarios using the Yo'LLaVA Nguyen et al. (2024) and This-Is-My-Img benchmarks. For completeness, we also compare against MyVLM on its dataset using CLIPScore Hessel et al. (2021) and Personalization Recall, as it lacks a VQA split.

PeKit consistently outperforms baselines on both single- and multi-concept VQA tasks. On the MyVLM dataset, PeKit surpasses MyVLM in generating image-aligned captions (CLIPScore) and correctly referencing personalized concepts (Personalization Recall). On the Yo'LLaVA benchmark, PeKit outperforms Yo'LLaVA without requiring any fine-tuning, special tokens, or model modifications.

On the This-Is-My-Img benchmark, Yo'LLaVA performs on par with the base LLaVA, while PeKit yields a 5% improvement, showing strong visual recognition and reasoning. In multi-concept cases, Yo'LLaVA degrades base model performance, whereas PeKit delivers a consistent 7% gain. Moreover, PeKit generalizes effectively to the video domain (3.5), unlike prior approaches, which are not readily extendable to video. PeKit improves the base model by 12% and maintains gains when combined with stronger instruction-following backbones.

In summary, our experiments demonstrate that PeKit consistently outperforms training-based personalization methods across diverse benchmarks, including single- and multi-concept VQA tasks. Its ability to

Table 5: VQA accuracy on **This-Is-My-Img Multi-Concept** and **This-Is-My-Video** datasets.

| This-Is-My-Img Multi-Concept VQA Accuracy | | This-Is-My-Video VQA Accuracy | |
|---|---|---|---|
| LLaVA | 49.0 | LLaVA-OneVision-Qwen2 | 23.0 |
| Yo'LLaVA | 12.7 | PeKit (LLaVA-OneVision-Qwen2) | 35.0 |
| PeKit (LLaVA) | 56.3 | InternVideoChat2.5 | 56.6 |
| PeKit (InternVL) | **63.6** | PeKit (InternVideoChat2.5) | **61.3** |

**Single-Concept**     **Multi-concept**     **Video**

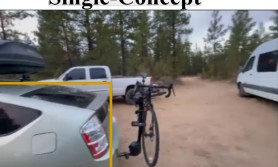 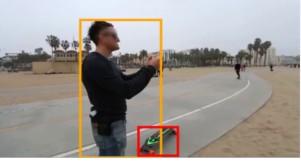 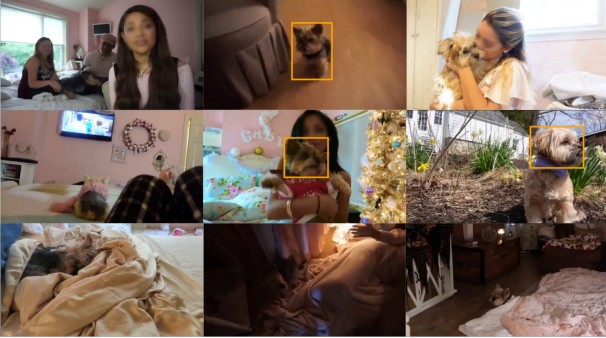

**Q:** Caption the image.
PeKit: "Nikki's car" is parked in a dirt parking lot, with a bicycle leaning against it. The car is positioned next to a truck, and there are trees in the background.

**Q:** What is Casey doing and where is Casey's boosted board placed?
PeKit: Casey is standing on a street, holding a cell phone and looking at it. Casey's boosted board is placed on the ground next to him.

**Q1:** In what activities is Lili involved? PeKit: Lili is involved in various activities such as being held by a person, standing on a wooden floor, and lying on a bed.
**Q2:** Is Lili wearing a specific item in any part of the video? PeKit: Yes, Lili is wearing a blue bandana in one part of the video. She is standing on a wooden floor with plants in the background.
**Q3:** Can you describe the scenes where Lili is being held up? PeKit: Yes. In the first scene, Lili is being held up by a woman in a bedroom. In the second scene, Lili is being held up by a woman in a bedroom with a Christmas tree in the background.

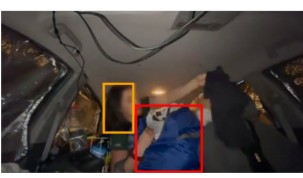

**Q:** Describe Casey's son's outfit.
PeKit: Casey's son is wearing a patriotic outfit, which includes a backpack and a shirt with stars on it.

**Q:** Where is Nikki's camper bag and what is Nikki doing?
PeKit: Nikki's camper bag is in the car, and Nikki is sitting in the back seat.

Figure 3: **Qualitative Results:** PeKit handles a range of personalization tasks, encompassing both single- and multi-concept personalization in images and videos. For video personalization, the VLM model can reliably track the target object across frames using only a few confidently annotated instances. One representative frame is shown per scene.

generalize without fine-tuning, maintain performance in multi-concept scenarios, and extend effectively to video settings highlights its robustness and scalability for real-world personalized visual understanding.

## 4.6 Qualitative Results

Fig. 3 presents examples of PeKit performing various tasks on the proposed benchmark, This-Is-My-Img. In the left column, PeKit successfully identifies personalized single-concept objects, even in ambiguous scenarios involving multiple instances from the same object category. In the middle column, it demonstrates the ability to answer questions involving multi-concept pairs. Lastly, without any modifications, the model can be applied to video inputs by visually prompting the model with detected personalized objects in the video. Further qualitative comparisons with the base LLaVA model, MyVLM, and Yo'LLaVA are presented in Appendices D.1, D.2, and D.3.

## 4.7 Ablation and Analysis

A comprehensive analysis of our method is provided in Appendix.A, where we analyze robustness under various settings. Specifically, we examine the effects of altering the backbone encoder $F_{emb}$, the number

of reference images per object ($N$), the number of personalized objects ($P$) in the dataset, the detection threshold ($\tau$) and the name ($n_p$) of the personalized objects.

In the following subsection, we additionally examine the robustness of our approach with respect to a requirement introduced by our framework, namely the semantic category $k_p$ assigned to each object (see Section 3.2).

### 4.7.1 Semantic Category $k_p$

As described in Section 3.2, we use the semantic category $k_p$ of each object in the reference image $I_p$ to derive its corresponding object-level mask $S_p$.

We note that, in practice, introducing an object for personalization typically involves specifying its name. In many cases, the name itself implicitly encodes the semantic category—e.g., 'My book' or 'Jack's car'—which can then be leveraged by an open-vocabulary segmentation model for view extraction.

Therefore, for the MyVLM dataset, where the object names are generic, we directly input them into our open-vocabulary detector. Similarly, on This-Is-My-Img dataset, aside from the people in the dataset (queried with the category 'Man/Woman'), all other object names in this dataset include their semantic category (e.g., Alex's hat, See Appendix.B).

For the Yo'LLaVA dataset, some concept names such as Vietnamese individual names do not directly indicate the semantic categories. We compare our method's performance on the Yo'LLaVA dataset by providing the open-vocabulary segmentation model with semantic categories, dataset-provided names, or the generic term 'main' during reference view extraction. As shown in Table 6, PeKit achieves state-of-the-art performance even without relying on semantic categories of personalized objects. Note that during inference we consistently use the term 'object' to extract object proposals using G-Dino.

Consequently, when the reference image for a personalized object is representative, with the subject clearly dominating the scene, our method remains competitive even without an explicitly provided semantic category $k_p$. If needed, a lightweight open-world classifier can be employed to infer this label prior to processing the object within our pipeline.

Table 6: Reference view vocabulary ablation on Yo'LLaVA dataset. PeKit achieves SOTA results even using the generic category 'main' to extract the personalized objects from the reference views.

| Method/Metric | Precision | Positive Acc | Negative Acc | Weighted Acc |
|---|---|---|---|---|
| Yo' LLaVA | - | **94.9** | 89.8 | 92.4 |
| PeKit (categories) | **74.8** | 91 | 98.7 | **94.9** |
| PeKit ('main') | 72.9 | 86.9 | **98.9** | 92.9 |
| PeKit (Vietnamese names) | 73.8 | 88 | **98.9** | 93.45 |

## 5 Limitations and Future Work

### 5.1 Noisy Reference Views

Our method relies on instance masks generated by a segmentation network. Inaccurate or noisy masks can lead to false positives during inference. As shown in Figure 4, poor masks (highlighted in red) result in incorrect matches (red bounding boxes). Applying clustering to reference features may help filter out erroneous views and improve matching reliability.

### 5.2 Small Object Representation

Due to the stride factor ($=14$) in the DINOv2 encoder, feature extraction occurs at a lower resolution than the original image. This limits detail capture for small objects, as illustrated in Figure 5, where only coarse features (e.g., color or category) are encoded, increasing false positives. A potential solution is to crop and resize small objects to DINOv2's native resolution ($518 \times 518$) before embedding.

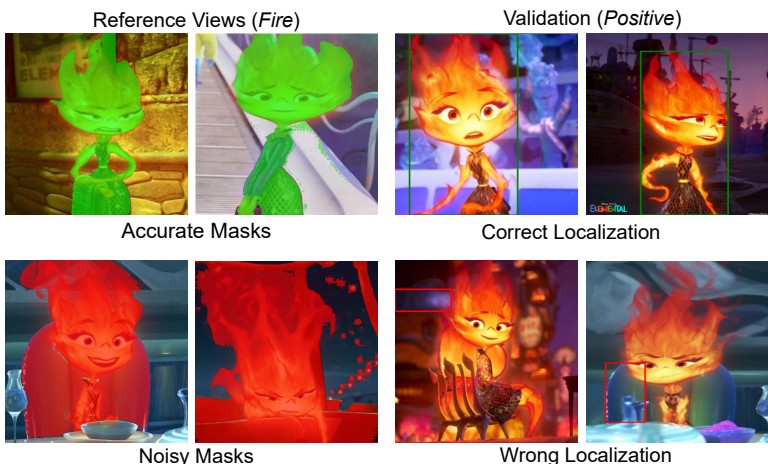

Figure 4: Noisy reference views: Poor segmentation masks may affect the visual prompting stage and degrade PeKit's performance.

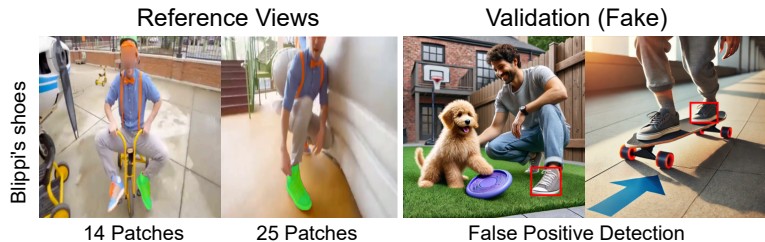

Figure 5: Small reference objects: Due to DINO's fixed patch size of $14 \times 14$, the resulting embeddings are relatively coarse, which can lead to increased false positives, especially for small or fine-grained objects.

### 5.3 Contextual Object Relationships

Our instance-level detection approach could benefit from contextual cues. For example, identifying *Alex's everyday bag* is more reliable when *Alex* is present in the scene. Future work will explore extracting object relationships from reference images and integrating them into the LVLM as prior knowledge.

## 6 Discussion and Conclusion

In this work, we introduced PeKit, a training-free, plug-and-play toolkit for LVLM personalization that combines vision foundation models with retrieval-augmented generation and visual prompting. Our approach outperforms existing training-based methods without requiring any fine-tuning or additional data beyond the personalized concept inputs. To evaluate our method, we proposed a challenging new benchmark that reflects more realistic scenarios involving multi-concept and video personalization, significantly increasing the complexity of the visual recognition task. PeKit demonstrates strong robustness and scalability across both multi-concept and video settings. Our benchmark reveals strong performance and shows some improvement points, particularly in handling complex temporal and multi-concept reasoning. We believe that PeKit establishes a new efficient training-free baseline for future research in LVLM personalization.

## 7 Ethical Considerations

PeKit is a training-free personalization approach for LVLMs designed to assist users in everyday tasks. It enables local and efficient customization without the need for retraining, supporting applications such as personalized robotics and visual assistants (see Figure 1 and Figure 10 Appendix).

We recognize potential risks if the method were misused for surveillance or applied without user consent. To address these concerns, PeKit operates entirely on-device, never transmits data externally, and works on extracted features rather than storing raw images. Personalization remains fully user-controlled, requiring explicit input.

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
