# Personalization Toolkit: Training Free Personalization of Large Vision Language Models (Appendix)

**Soroush Seifi**[*] **Vaggelis Dorovatas**[*] **Matteo Cassinelli**[*] **Fabien Despinoy** **Daniel Olmeda Reino** **Rahaf Aljundi**

*Toyota Motor Europe*

**Reviewed on OpenReview:** *https://openreview.net/forum?id=5mbn3B0O29*

## A   Ablation

In this section we ablate various aspects of our personalization method primarily using the Yo'LLaVA dataset as it serves as a well-established benchmark.

### A.1   Retrieval Module Backbone

We present an ablation study of the image encoder $F_{emb}$, which serves as the backbone of the Retrieval Module and is responsible for extracting object features and performing matching. Table 1 shows a noticeable gap between CLIP and DINO, indicating that DINO's visual features are more discriminative and better suited for retrieval tasks. Between the two DINO variants (base and large), the larger version delivers slightly better performance, the difference is marginal indicating that any DINOv2 backbone can be deployed.

Table 1: Feature extractor ablation. DINO features significantly outperform CLIP features.

| Yo' LLaVA Dataset | | | | |
|---|---|---|---|---|
| Method/Metric | Precision | Recall | | |
| | | Positive | Negative | Weighted |
| DINOv2 (Large) | 74.8 | **91** | **98.7** | **94.9** |
| DINOv2 (Base) | **75.5** | 90.8 | **98.7** | 94.7 |
| CLIP (Large) | 69 | 78.3 | 98.2 | 88.3 |

### A.2   Number of Reference Images (*N*)

Since our approach does not require a training phase, a key question is how many reference images are needed for robust visual recognition of personalized objects. Figure 1 shows that our method performs well with just one reference image and matches state-of-the-art performance with two images on MyVLM dataset. On Yo'LLaVa dataset, we achieve comparable performance to Yo'LLaVA Nguyen et al. (2024) with only three images, even though the full set includes up to 10 images for some objects.

### A.3   Number of Personalized Objects (*P*)

Another key consideration for any personalization method is its robustness to increasing number of personalized objects, particularly as more intra-category instances are introduced into the dataset. Figure 2 illustrates the performance of PeKit on the first 10 objects from the Yo'LLaVA dataset as the number of personalized objects increases incrementally from 10 to all 40 categories. While there is a slight performance drop at higher values of P, PeKit demonstrates overall stability and robustness.

---

*[*]Providing contracted services at Toyota Motor Europe.

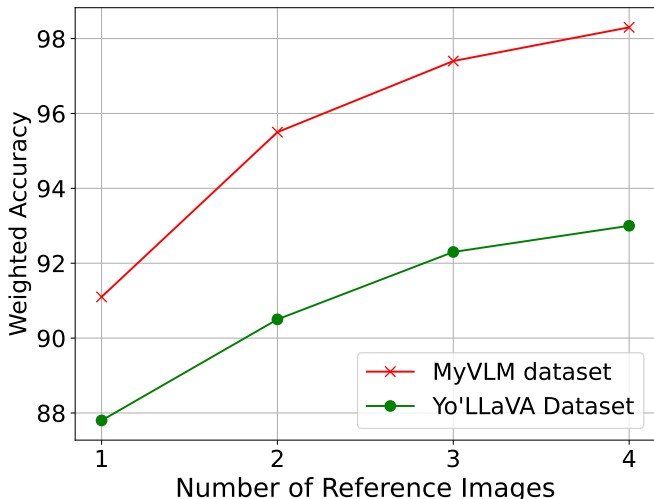

Figure 1: Ablation on N: Average weighted visual recognition accuracy as a function of number of reference images. Increasing the number of reference images improves performance, but PeKit is robust with just one reference image.

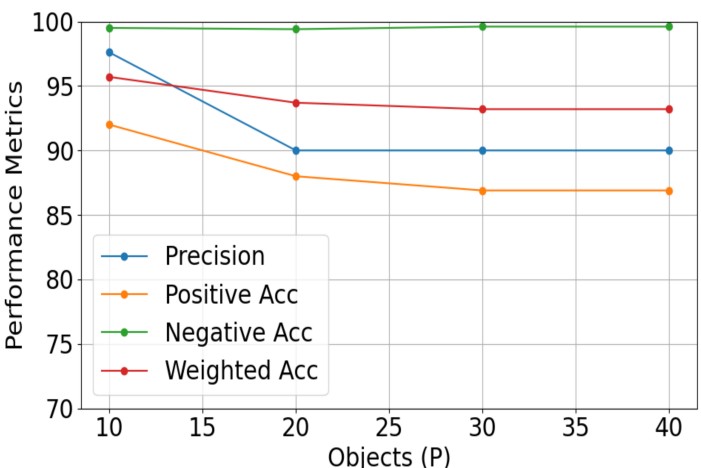

Figure 2: Ablation on P: PeKit's performance when evaluated on a subset of 10 categories from Yo'LLaVA dataset while progressively increasing number of objects.

### A.4 Threshold Selection

In the main paper, we employ a fixed threshold of $\tau = 0.75$ across all experimental settings to determine whether a personalized object is present in an image.

Figures 3 and 4 illustrate the precision–recall trade-off for different values of $\tau$ on the Yo'LLaVA dataset. As shown, $\tau = 0.75$ achieves the highest F1-score, providing the best balance between precision and recall.

Lowering the threshold increases the number of detections but also raises the likelihood of false positives. Conversely, increasing the threshold improves precision at the cost of reduced recall, potentially causing some objects to be overlooked.

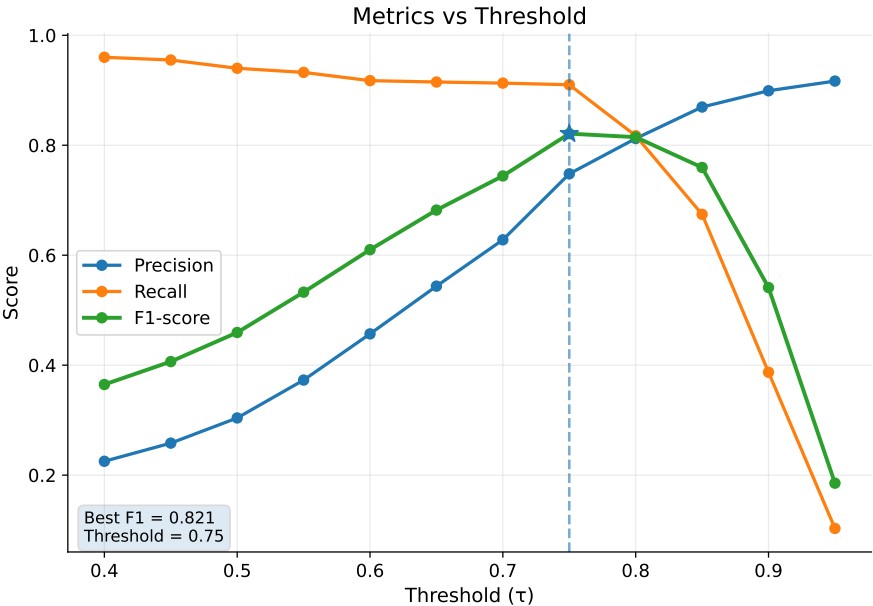

Figure 3: Precision, Recall and F1 score curve for different detection thresholds on Yo'LLaVA dataset. The best performing threshold (0.75) in terms of F1 score is marked with a star.

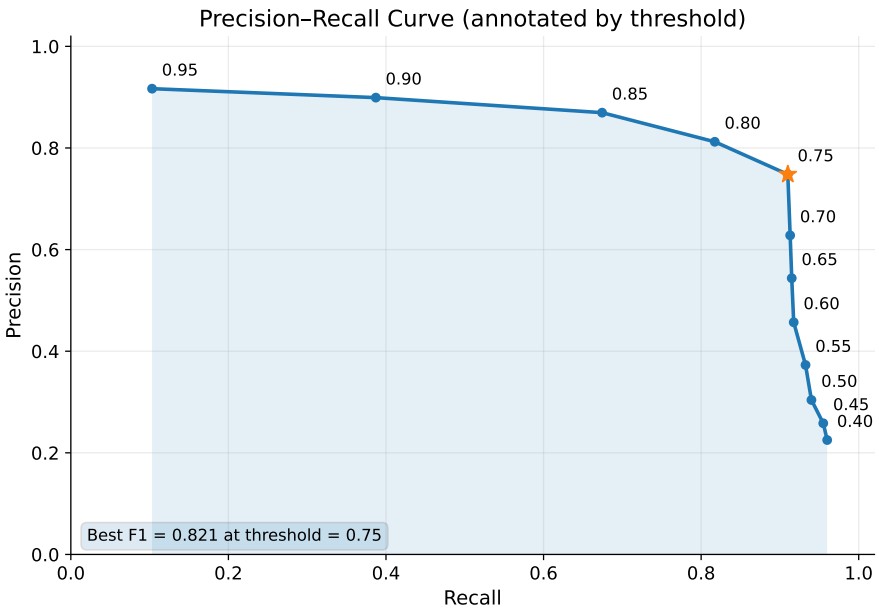

Figure 4: PR-curve on Yo'LLaVA dataset. The threshold achieving the best F1 score is marked with a star.

For instance, in a personalized search task such as the query "find my keys," a high threshold may prevent the model from detecting the keys altogether, whereas a low threshold may return multiple key candidates, allowing the user to manually identify the correct one. In this study, we selected the threshold 0.75 that best balances precision and recall on the Yo'LLaVA dataset and applied it across all experiments, while noting that the threshold may be adjusted depending on the requirements of a specific application.

For completeness and flexibility, we further provide a straightforward method for tuning the thresholds on a per-object basis. To adjust the threshold for a given personalized object $i$, we first compute the distribution of distances for the reference images of the object, and then calculate the mean $\mu_i$ and standard deviation $\sigma_i$

of the distribution. The threshold is then set as $\tau_i = \mu_i - \sigma_i/2$. We note that by using this method, one can easily adjust the thresholds to be stricter or more loose by adding a scalar multiplier, as $\tau_i = \mu_i - \alpha(\sigma_i/2)$. Table 2 shows that this method can produce even higher accuracy.

Table 2: Automatic object threshold selection.

| Yo' LLaVA Dataset | | | | |
|---|---|---|---|---|
| Method/Metric | Precision | Recall | | |
| | | Positive | Negative | Weighted |
| Fixed $\tau = 0.75$ | **74.8** | 91 | **98.7** | 94.9 |
| Tuned $\tau = \mu - \sigma/2$ | 66 | **92.5** | 98.1 | **95.3** |

### A.5 Video-QA Object Naming

Object names often offer important contextual clues about the subject in a query to the base LVLM. For example, when asked 'What is Jack doing?', the model may infer that 'Jack' refers to a male individual in the image. In such cases, the LVLM can often respond accurately without the need for personalization. To isolate the effect of personalization, we perform an ablation study on an open-ended video QA task, where all identified personalized objects are visually prompted using the generic label ENTITY (see Appendix C and Fig.6 for prompt templates). Table 3 compares our method's performance when using object names or the term ENTITY to the base LVLM models.

### A.6 Time and Memory Requirements

Table 4 details the memory usage and runtime of the various modules in our method for the LVLM personalization task. By default, we employ GroundingDINO (Base) Liu et al. (2023b) and SAM (Base) Kirillov et al. (2023) for extracting views from reference images and proposing objects during inference, DINOv2 (Large) features for retrieving personalized instances, and LLaVA 1.5 13B as our LVLM for generating answers. All experiments were conducted using an A5000 RTX GPU. As shown in the table, our plug-and-play modules are efficient, adding minimal overhead to the original LLaVA model.

Specifically, our approach introduces an overhead of 0.35 seconds per image (0.31 seconds from G-DINO) relative to LVLM's inference time. This overhead can be substantially reduced by employing a lightweight object proposal network such as YOLO-World Cheng et al. (2024), which runs at 55 FPS compared to G-DINO's 3 FPS. However, as reducing computational time is not the primary focus of this work, we leave further optimization in this direction for future research.

## B This-Is-My-Img Benchmark Details

The original This-Is-My dataset Yeh et al. (2023), designed for video-level detection of personalized objects, comprises 104 training and 582 validation short video segments spanning 15 categories. However, some of the segments are no longer available for download. Consequently, one personalized object category, 'Alex's Piano' has been removed from our proposed benchmark due to the unavailability of its training segments.

Table 3: Video-QA naming bias ablation. Our method mitigates naming bias, improving baseline VLM accuracy in ambiguous settings without explicit object names.

| Model/Variation | Base | PeKit | Base+ENTITY | PeKit +ENTITY |
|---|---|---|---|---|
| **LLaVA-OneVision** | 23.0 | **35.0** | 14.0 | 19.9 |
| **InternVideoChat** | 56.6 | **61.3** | 45.8 | 56.2 |

Table 4: Time and memory requirement of PeKit. View extraction is conducted only on the reference views corresponding to each personalized object category, whereas the remaining modules are executed for every image during inference.

| Module | Backbone | Time/Image (S) | Memory (GB) |
|---|---|---|---|
| View Extraction | G-DINO + SAM | 0.48 | 1.2 |
| Object Proposal | G-DINO | 0.31 | 0.87 |
| Retrieval | DINOv2 | 0.04 | 1.8 |
| Reasoning | LLaVA 1.5 13B | 0.87 | 16.8 |

## B.1  Single-concept Set

Table 5 presents further details about the personalized objects and the number of frames included in the single-concept validation splits of our benchmark. We sampled every 10th frame of each validation video segment and manually assigned frames to the respective splits. Note that three categories lack a **Negative (Hard)** set because the corresponding objects are present in every frame of their validation segments. The number of **Positive**, **Negative (Hard)**, and **Negative (Other)** frames varies between categories due to differences in segment length and the number of segments per object. In contrast, the **Fake** set was standardized, with 10 validation frames generated for each class. Figure 5 offers additional visual examples from our benchmark.

The single-concept validation also includes a VQA set comprising 70 images (5 per category). Challenging frames were manually selected, and initial question–answer pairs were generated using the GPT-4O model. These pairs were then manually refined to create a high-quality evaluation set. In line with the Yo'LLaVA framework, answers for this set are presented in an A/B multiple-choice format.

Table 5: Single-concept categories and number of frames of This-Is-My-Img benchmark. The validation frames in the dataset are organized into three subsets for each category: frames where the object is visible (*Positive*), frames from the same video segments where the object is absent (*Hard*), and positive frames belonging to other object categories (*Other*). In addition, the benchmark provides a GPT-generated set of 10 images per category (*Fake*).

| Category | Positive | Negative (Hard) | Negative (Other) |
|---|---|---|---|
| Alex's Bag | 161 | 43 | 2096 |
| Alex's Hat | 120 | 0 | 2137 |
| Blippi's Shoes | 339 | 79 | 1918 |
| Casey's Boosted Board | 35 | 118 | 2222 |
| Casey's Friend Marlan | 24 | 4 | 2233 |
| Casey's Son | 46 | 15 | 2211 |
| Gab's Puppy Lili | 56 | 12 | 2201 |
| Nikki's Camper Bag | 229 | 76 | 2028 |
| Nikki's Car | 651 | 184 | 1606 |
| Reynard's Keyboard | 162 | 29 | 2095 |
| Reynard's Work Chair | 188 | 59 | 2069 |
| Sherry's Road Bike | 95 | 12 | 2162 |
| Zak's Dog Coffee | 26 | 0 | 2231 |
| Zak's Dog Kona | 125 | 0 | 2132 |
| Sum | 2257 | 631 | 29341 |

## B.2  Multi-concept Set

Table 6 outlines the category pairs used in our multi-concept validation benchmark. This benchmark comprises 55 images, with each category pair represented by 5 manually selected positive examples sourced from the validation frames. We extended the original This-Is-My-Img dataset categories (Table 5) by incorpo-

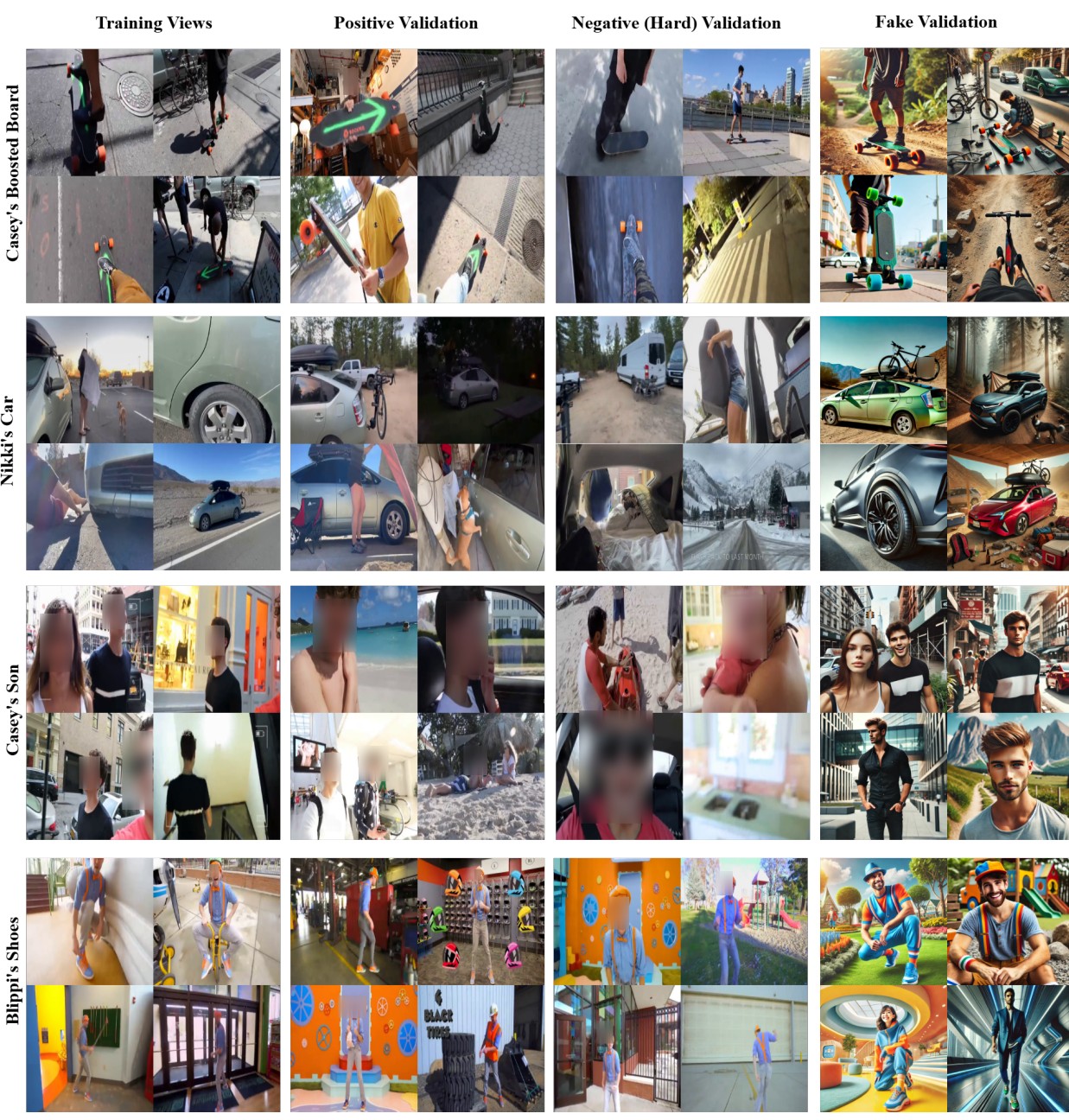

Figure 5: This-Is-My-Img single-concept benchmark. Our benchmark includes a wide range of concepts presented in realistic indoor and outdoor environments. Reference views can occasionally be sub-optimal, which increases the difficulty of the task. The positive validation set may contain false positives from within the same semantic category, allowing us to assess a model's robustness to contextual similarities. The negative (hard) and Negative (fake) validation sets are crafted to challenge model resilience by imitating the appearance of personalized objects or their surrounding context. Additionally, the negative ('other') set—though not shown in this image—includes images of all other personalized objects for each concept, serving to evaluate the method's robustness to dataset bias.

rating new personalized categories: Alex, Blippi, Casey, Gab, Nikki, Sherry, and Zak. For each of these added categories, we selected 5 reference views from the training segments. Each image in the benchmark is accompanied by an open-ended question–answer pair, collaboratively crafted with GPT-4O. This open-ended visual question answering (VQA) format raises the task's difficulty and reduces the likelihood that the underlying LVLM model can succeed through answer-choice elimination alone.

Table 6: Multi-concept categories on This-Is-My-Img benchmark. Each category-pair comes with 5 positive frames and 50 negative frames from other category pairs.

| Category-pair | |
|---|---|
| Alex - Alex's Bag | Nikki - Nikki's Car |
| Alex - Alex's Hat | Nikki - Nikki's Camper Bag |
| Blippi - Blippi's Shoes | Sherry - Sherry's Road Bike |
| Casey - Casey's Boosted Board | Zak - Zak's Dog Coffee |
| Casey - Casey's Son | Zak - Zak's Dog Kona |
| Gab - Gab's Puppy Lili | |

### B.3 Video-QA Set

As shown in the main paper, PeKit can be extended to video personalization by applying the method to sampled frames. To evaluate this, we used validation video segments (typically under 15 seconds) and curated a VQA dataset with 267 high-quality question–answer pairs for the single-concept categories in Table 5. For each video segment, we employed the LLaVA-OneVision-Qwen2-7B model with a 1-in-16 frame sampling rate to generate an initial set of 1,380 question–answer candidates. These were then filtered for duplicates using GPT-3.5 Turbo, reducing the set to 618 pairs. Finally, we manually refined the results to produce a curated set of 267 high-quality QA pairs.

## C Prompt Templates

In this section, we present the specific query format used across different parts of our experiments.

### C.1 Visual prompting

As mentioned in the main paper (Sec 3.4), we employ visual prompting to indicate the personalized object's location in the image. Next we query our LVLM to personalize its answer given the instance's name and context. Our prompt to the LVLM for various tasks follows this general structure:

> In this image (video), the entity enclosed in a '**COLOR**' box is called '**NAME**'.
>
> Without mentioning the bounding box and its color, '**TASK**'.
>
> [Optional] Give more details using the information from '**CONTEXT**'.

The **COLOR** placeholder indicates the color of the bounding box overlaid on the image, the **NAME** placeholder specifies the instance name, and the **TASK** placeholder contains the task-specific query. Optionally, the **CONTEXT** placeholder can contain the prior knowledge about the personalized object retrieved from the memory module.

Note that when multiple objects are detected in one image, the query's grammatical structure changes to a plural format, and the **COLOR** and **NAME** placeholders will contain multiple values separated by commas. The **CONTEXT** placeholder for each object is included in angle brackets ($<>$) and contains the **NAME** for the corresponding instance.

For the experiments in the paper the **TASK** placeholder can be any of the following prompts:

> **Personalized Captioning**: Describe what is '**NAME**' doing. Describe the image too.

**VQA**: Answer the following question about '**NAME**': '**QUESTION**'.

Figure 6 illustrates an example of our full prompt for each one of the tasks.

Figure 6: Prompt format. Personalized VQA and captioning on Yo'LLaVA (Left) and MyVLM (Right) datasets. The context used for the 'red chicken' is imaginary and generated by ChatGPT.

## C.2 Open-ended VQA Validation

To evaluate the accuracy of model predictions in an open-ended VQA task, we adopt the evaluation pipeline proposed by Maaz et al. (2023). In particular, we utilize the following prompt template to query a GPT-3.5-Turbo model for assessing the semantic alignment between predicted answers and ground truth responses for each question in the VQA dataset.

'You are an intelligent chatbot designed for evaluating the correctness of generative outputs for question-answer pairs. Your task is to compare the predicted answer with the correct answer and determine if they match meaningfully.

Here's how you can accomplish the task: INSTRUCTIONS:

- Focus on the meaningful match between the predicted answer and the correct answer.

- Consider synonyms or paraphrases as valid matches.

- Evaluate the correctness of the prediction compared to the answer.

Please evaluate the following video-based question-answer pair:

Question: **QUESTION**

Correct Answer: **ANSWER**

Predicted Answer: **PREDICTION**

Provide your evaluation only as a yes/no answer. Please generate the response in the form of a Python dictionary string with key 'pred', where value of 'pred' is a string of 'yes' or 'no'.

DO NOT PROVIDE ANY OTHER OUTPUT TEXT OR EXPLANATION. Only provide the Python dictionary string. For example, your response should look like this: 'pred': 'yes'.'

We calculate the VQA accuracy by dividing the number of times the GPT model, using the specified prompt template, responds with 'yes' by the total number of question-answer pairs in the set.

## D Qualitative Results

### D.1 Qualitative Comparison to LLaVA

Figure 7 presents examples of PeKit model compared to the base LLaVA Liu et al. (2023a) on VQA and personalized captioning tasks using images from all three benchmarks discussed in the main paper. When

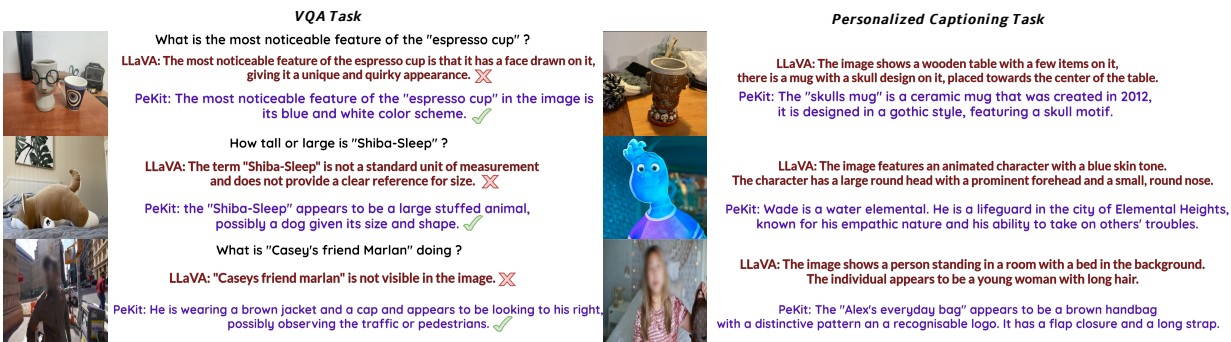

Figure 7: Qualitative comparison to LLaVA. Right: Our method detects personalized objects and integrates provided context (for qualitative comparison) in caption generation. Left: While the original model struggles with specific questions about named objects, our method easily identifies the referred object.

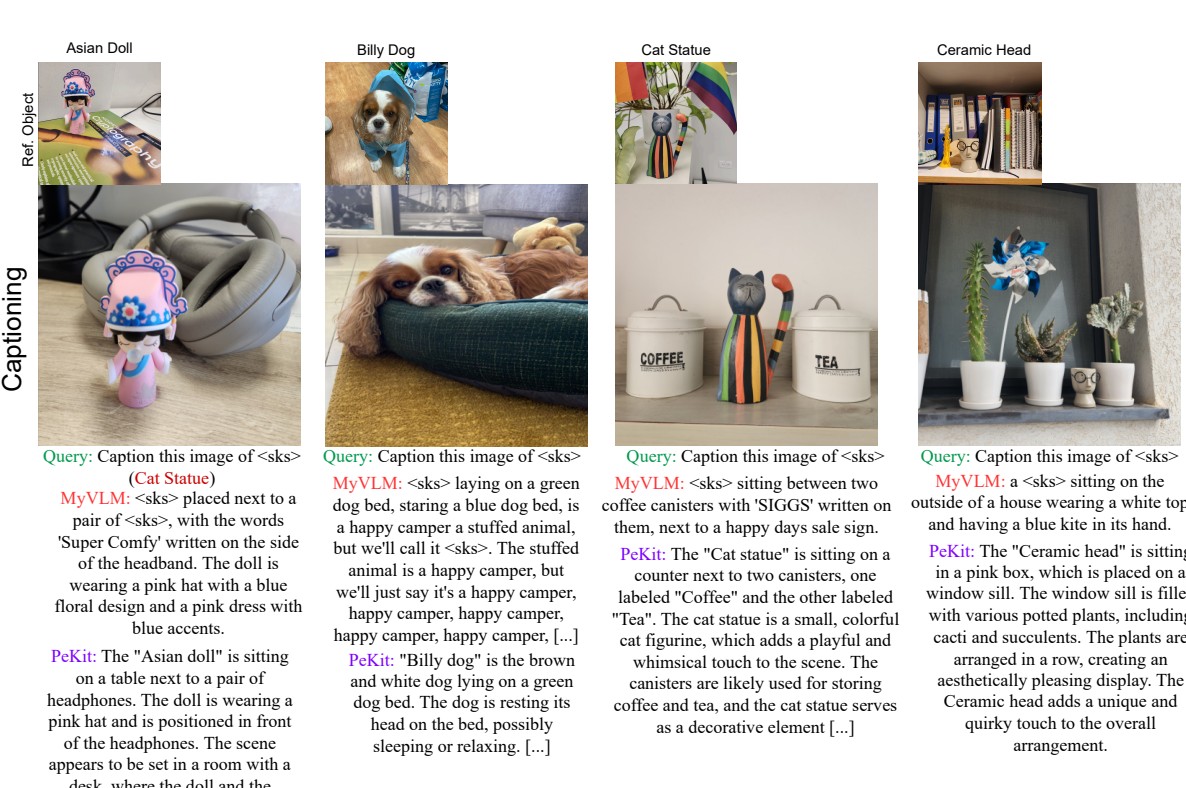

Figure 8: Qualitative comparison to MyVLM. MyVLM often misidentifies personalized objects because of its low precision. In the leftmost figure, when prompted to caption an image containing a 'Cat Statue'—which is actually absent—MyVLM incorrectly labels the 'Asian doll' and the headset as the 'Cat Statue' instead of rejecting the query. Additionally, MyVLM training interferes with the original captioning capabilities of the LVLM, leading to hallucinations, short captions, and sometimes incomprehensible text. For each image, 'Query' depicts MyVLM's sytem prompt where the concept identifier <sks> is replaced with the personalized object's name. PeKit employs its own prompt template described in Appendix C.1.

LLaVA does not recognize an object from the given name in the query, it makes guesses, leading to hallucinations or incorrect statements. In contrast, PeKit accurately identifies objects and uses in-context information to guide LLaVA in answering questions or providing details about the image, effectively incorporating

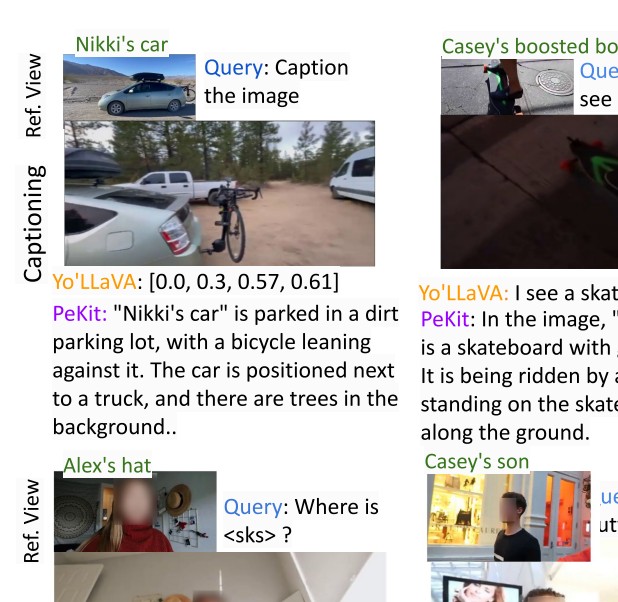

Figure 9: Qualitative comparison to Yo'LLaVA. Yo'LLaVA's prompt template requires specifying the personalized object's identifier in the query (first row), limiting generalization since users must already know which objects are in the image. Using image-level embeddings can also cause confusion between similar objects (e.g., Alex vs. Alex's bag). Adjusting the LLM's head weights further harms captioning quality. PeKit achieves better captioning quality without any training. In each example, the 'Query' shows Yo'LLaVA's prompt with the concept identifier replaced by the object's name and an added system prompt. PeKit uses a different template, detailed in the Appendix C1.

in-context information and object appearance. While more advanced prompting or personalized response examples could enhance PeKit, we opted for simplicity and standard design, leaving such improvements for future work.

## D.2 Qualitative Comparison to MyVLM Alaluf et al. (2024)

Figure 8 compares PeKit with MyVLM on images from the MyVLM dataset. We used the original checkpoints and code provided by the authors of MyVLM to generate the results. Checkpoints for the VQA task were not provided.

MyVLM shares the same limitation as Yo'LLaVA, functioning exclusively when the concept identifier is included in the query. Additionally, MyVLM exhibits low precision in detecting personalized objects, as shown in the main paper for the This-Is-My-Img benchmark. This can lead to misidentifying objects as personalized ones. The leftmost image in Figure 8 demonstrates this limitation. MyVLM is asked to provide a caption for the target concept 'Cat Statue' while the provided image includes another personalized object, 'Asian Doll.' As shown, MyVLM incorrectly identifies the 'Asian Doll' as the 'Cat Statue' and generates an incorrect personalized caption. Our method addresses this issue by first detecting the correct personalized object(s) and then generating a caption based on the prompt template provided in Appendix C.1.

Furthermore, MyVLM's training appears to degrade the original LVLM's captioning capabilities, leading to short captions with hallucinations and sometimes incomprehensible text, leading to a low CLIPScore as demonstrated in the main paper.

## D.3 Qualitative Comparison to Yo'LLaVA Nguyen et al. (2024)

Figure 9 compares PeKit to Yo'LLaVA for VQA and personalized captioning tasks on This-Is-My-Img benchmark. As seen on the first row, Yo'LLaVA's prompt template requires the query to include the target object's concept identifier, making it unsuitable for general captioning tasks and tailored for Visual Question Answering. Besides, since Yo'LLaVA operates on image-level embeddings of reference views, it needs clutter-free object-centered reference views of the personalized objects. As seen on the second row, performance can decline if the personalized object is not in the foreground or if there are other objects/people interacting with the personalized object in the reference views. Besides, fine-tuning the last layer of the language model reduces the LVLM's captioning capabilities for Yo'LLaVA.

## D.4 Real-world Demonstration

To evaluate PeKit in real-world conditions, we deploy it on a mobile manipulation robot for a "search and fetch" task (Figure 10). The robot autonomously explores an unfamiliar environment to locate and grasp a specific object, in this case a carton milk box named 'My Milk'.

The robot, equipped with a wheeled base and a single-arm manipulator, combines mobility and dexterity for effective navigation and interaction. Exploration is guided by MORE Mohammadi et al. (2025), a pipeline that uses scene graphs and large language models (LLMs) to plan actions from a 3D panoptic map generated by OpenVox Deng et al. (2025). We enhance OpenVox with occupancy mapping and integrate PeKit to refine object labels for accurate identification.

Before deployment, PeKit's RAG memory is initialized using a small set of target object images processed through our view extraction pipeline. Refined labels and the panoptic map are shared with MORE via ROS. To grasp the object, the robot navigates to its location, captures an RGB-D image, extracts the object mask using projected panoptic data refined by SAM Kirillov et al. (2023), and estimates a grasping pose via a rule-based method. This enables successful retrieval of specific items—e.g., a chosen milk carton from a shelf with similar packages.

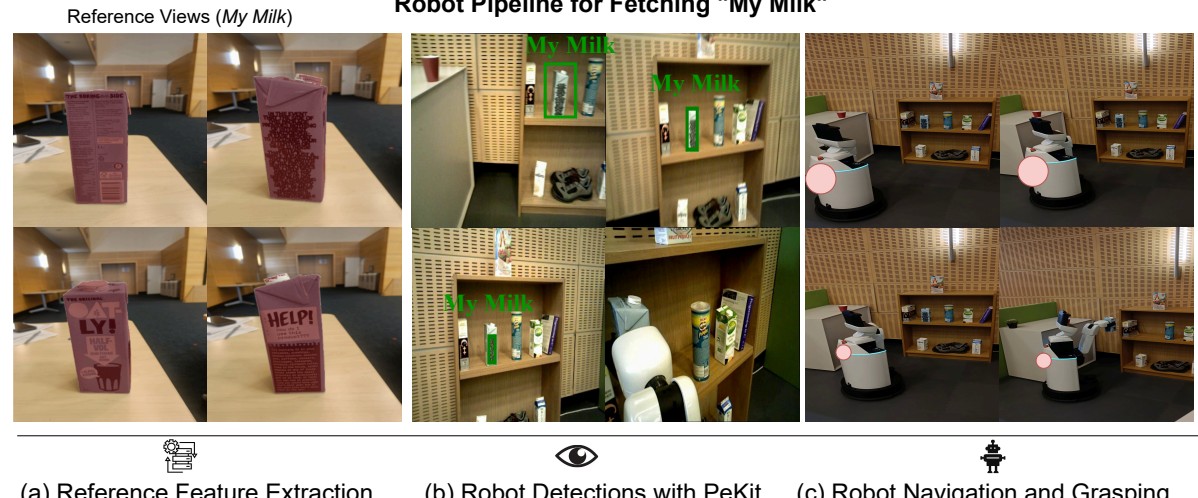

Figure 10: Real-world demonstration. PeKit can be incorporated into a mobile manipulation robot to perform personalized object search and fetching.