# OpenReview forum: "Personalization Toolkit: Training Free Personalization of Large Vision Language Models"
_TMLR — Accepted by TMLR_

### Review · Reviewer_1yAj · 2025-11-20

**Summary Of Contributions:**

This paper tackles the problem of personalizing Large Vision-Language Models (LVLMs) so that they can recognize specific users or object instances and answer personalized queries about them, without any additional training.
The authors propose PeKit, a modular, plug-and-play personalization toolkit that uses open-vocabulary segmentation (GroundedSAM / GroundingDINO), DINOv2 embeddings, a memory module, and retrieval-augmented visual prompting to personalize off-the-shelf LVLMs with only a few reference images per concept, without any fine-tuning.
The pipeline naturally extends from single personalized objects to multi-concept images and to videos, by tracking multiple personalized instances and visually prompting video LVLMs.
The paper introduces a benchmark built from the This-Is-My dataset, with single-concept, multi-concept, and video QA splits. The benchmark contains challenging negatives (same scene without object, other personalized instances, and GPT-generated “fake” images) to better reflect realistic personalization scenarios.
PeKit achieves state-of-the-art results on existing MyVLM and Yo’LLaVA benchmarks and on the new benchmark, across both recognition and VQA metrics, while remaining training-free and model-agnostic

**Audience:**

Yes

**Audience Explanation:**

The work sits at the intersection of LVLMs, personalization, retrieval-augmented generation, and video understanding. It addresses a practical and increasingly important question: how to personalize strong LVLM backbones for user- and instance-specific reasoning without expensive per-concept training, and it introduces a challenging benchmark that highlights limitations of current approaches.

**Broader Impact Concerns:**

The work enables personalized recognition of specific people and objects in images and videos, and demonstrates that this can be done efficiently and training-free. This clearly has potential positive impact for assistive technologies and user-centric visual assistants, but it also raises privacy and surveillance concerns:
The ability to track named individuals or possessions across cluttered, real-world scenes and videos could be misused for surveillance, stalking, or targeted profiling if deployed without proper consent and access control.
The benchmark is based on a real-world dataset of people and their belongings; although the authors blur faces and mention GDPR compliance in their figures, there is no dedicated broader-impact or ethics section explicitly discussing potential misuse, data subject consent, or recommended safeguards (e.g., on-device processing, opt-out mechanisms).

**Claims And Evidence:**

Yes

**Claims Explanation:**

The core claims are that (1) LVLM personalization can be done in a training-free, plug-and-play way; (2) the proposed method scales to multi-concept and video scenarios; (3) the new benchmark is more realistic than prior ones; and (4) PeKit achieves state-of-the-art performance on existing and new personalization benchmarks

These claims are largely backed by:

Quantitative results showing consistent gains over MyVLM and Yo’LLaVA on their own benchmarks and on the new This-Is-My-Img / This-Is-My-Video settings, across recognition and VQA metrics.

Ablation studies on the choice of encoder (DINO vs CLIP), the number of reference views, the number of personalized objects, and thresholding strategies, which support the design choices and show robustness to scaling the number of concepts.

Detailed benchmark description including labeled positive and multiple types of negative samples (hard, other, fake) and clearly defined metrics (precision, positive/negative/weighted accuracies, CLIPScore, personalization recall, VQA accuracy).

**Requested Changes:**

Expand discussion of dataset and benchmark release. The benchmark is described as “previewed” and to be made publicly available. Providing a clearer statement on planned release (code, data splits, evaluation scripts) would significantly help reproducibility and adoption.

---

> ### Author Response · Authors · 2026-02-01
>
> - We would like to thank the reviewer for their constructive feedback.
> - We have uploaded the full benchmark on Google Drive and shared the link on the revision of the paper.
> Benchmark is available at: https://drive.google.com/drive/folders/1r13Si4PLlEXnCHQlwUMJEyHpFyQYgSrj?usp=sharing
> - We have initiated a GitHub repository for this project, which is expected to be made public by the camera-ready deadline.
> - Our manuscript includes a dedicated discussion (Section 7, Ethical Considerations) on ethical considerations. In summary, our method operates entirely on-device and stores extracted features rather than raw images. As a result, it remains fully user-controlled and requires explicit user input for personalization.

---

> > ### Comment · Reviewer_1yAj · 2026-02-02
> >
> > Thanks for your feedback. I hope the author can make the github repo when this paper camera-ready. With that, my concerns have been resolved, and I support acceptance of the paper.

---

### Review · Reviewer_M1cu · 2025-12-05

**Summary Of Contributions:**

**Summary**

PeKit introduces a training-free approach for personalizing LVLMs to recognize specific object instances. It uses DINOv2 features, RAG-based retrieval, and visual prompting to guide model outputs without fine-tuning. The paper contributes This-Is-My-Img, a benchmark with multi-concept scenarios, video inputs, and challenging negatives. PeKit achieves SOTA on existing benchmarks while demonstrating strong precision and scalability.

**Strength**

*Practical Impact*: Training-free approach enables scalable deployment, addressing the critical limitation of prior work requiring per-object fine-tuning.

*Realistic Benchmark*: This-Is-My-Img includes hard negatives (background scenes, fake objects), multi-concept scenarios, video evaluation, and cluttered conditions, which are significantly more challenging than prior benchmarks.


**Weakness**

*Recall/Precision Trade-off Inadequately Addressed:* PeKit achieves only 69.0% recall compared to Yo'LLaVA's 87.1% on This-Is-My-Img positive images, which miss nearly one-third of target objects. While the paper emphasizes precision gains, many applications (e.g., "find my keys") require high recall. Critical missing elements: 1) No precision-recall curves showing the threshold trade-off. 2) Fixed τ=0.75 across all datasets lacks justification or sensitivity analysis

*Semantic Category Requirement Undermines "Training-Free" Claim:* The method requires a semantic category, kp for segmentation (Eq. 1), which represents a significant practical limitation, casually mentioned but inadequately addressed. How do users specify categories for novel or ambiguous objects? The claim that the method achieves "SOTA even without it" (using the generic 'main' category) is relegated to Appendix A.5 without a main-text performance comparison. This undermines the core contribution if users must provide careful category labels or accept degraded performance. Clear guidance on category specification and performance impact is needed.

**Audience:**

Yes

**Audience Explanation:**

The paper addresses timely problems in LVLM personalization with a practical training-free approach, introduces a more realistic benchmark, and demonstrates applicability across multiple models and settings

**Claims And Evidence:**

Yes

**Claims Explanation:**

Core technical claims are supported by diverse experiments

**Requested Changes:**

Please refer to the Weakness.

---

> ### Author Response · Authors · 2026-02-01
>
> - We would like to thank the reviewer for their constructive feedback.
> -  We have added a precision–recall trade-off analysis to the revised supplementary material (Appendix A.4, Threshold Selection). We have selected the threshold with the best trade-off on Yo'LLaVA dataset as our fixed threshold across all experiments. For the query suggested by the reviewer “find my keys,” a high threshold may prevent the model from detecting the keys altogether, whereas a low threshold may return multiple key candidates, allowing the user to manually identify the correct one. To address this, Appendix A.4 also introduces a simple method for tuning thresholds on a per-object basis.
> - We have moved the ablation study on the semantic category parameter (Kp) to the main text and expanded the accompanying discussion. We note that all personalization datasets provide object names, which can often be used directly as semantic categories for reference view extraction. Besides, in cases where reference images depict the personalized concept as the main subject, even using the query “main” effectively guides the open-vocabulary segmentation module to correctly segment the target object with a minimal degradation of the performance.

---

### Review · Reviewer_Mg9F · 2026-01-18

**Summary Of Contributions:**

The paper introduces PeKit, a method for personalizing VLMs. This approach enables VLMs to work with specific instances of objects, rather than object categories, and therefore to perform VQA tasks that are instance-specific. PeKit is an instance-based method, in the sense that no additional training is required for the neural network models used. In fact, PeKit is also agnostic to the particular network models used for extracting image features or modeling language. This contrasts with other methods that require fine-tuning or test-time training of such network models. Moreover, the paper introduces This-Is-My-Img and This-Is-My-Video, two new benchmarks for this particular application. The set of results and ablation studies seems fairly complete.

**Additional Comments:**

None.

**Audience:**

Yes

**Audience Explanation:**

I would assume that researchers working in this area would want to know about this. Because it is an important subproblem of the more general functionality of VLMs. However, it is an important case because there are clear applications, for instance robotics, where the personalization functionality is very important.

The paper's approach is straightforward. At its core, it is a simple augmentation of current VLMs and a couple of additional computer vision tools with k-nearest neighbor techniques. So, this paper does not introduce new ML techniques. On the other hand, it establishes an important baseline. It is striking that this simple augmentation improves so much over training-based approaches. This tells us that the direction the community has been going so far regarding this problem is not the right one. I would expect that future developments should compare at least against this baseline to show that we are making progress.

**Broader Impact Concerns:**

I have explained above the importance of the paper.
Usually, I do not like papers that do not present a clear technical contribution. On the other hand, I recognize that it is important to make the community realize that for this problem, we can use current models and augment them with k-nearest neighbor techniques. So, we should focus on beating this baseline. This is an important point to ensure the community moves in the right direction.

**Claims And Evidence:**

Yes

**Claims Explanation:**

The paper is very clear and well written. I would also add that:
- The problem of personalization of VLM is relatively new and not very well investigated.
- The results are quite convincing, and they are a strong experimental validation that the approach has merit.

**Requested Changes:**

No particular requests. The paper is simple, clear, and timely.

---

> ### Author Response · Authors · 2026-02-01
>
> We would like to thank the reviewer for taking the time to review our manuscript and for their positive assessment of its clarity, timeliness, and simplicity.

---

### Decision · Action_Editor_NSVA · 2026-03-15

**Recommendation:** Accept as is

**Audience:**

Yes

**Audience Explanation:**

This paper addresses the topic of  personalisation of large vision language models, which is a very important, but relatively under-explored problem. With the wide applicability of these models, it will be of interest to several members in TMLR's audience.

**Claims And Evidence:**

Yes

**Claims Explanation:**

This paper addresses a very relevant, but relatively under explored problem of personalizing large vision language models, which can then recognise specific object instances. Towards this goal, they propose a training free approach, termed PeKit, which is in contrast to existing approaches which require costly training for each personalised item. The proposed model-agnostic toolkit allows personalisation of multiple concepts for both images and videos. The benchmark introduced in this work will also be useful to the research community and will help to advance research in this important area.

Extensive experiments are conducted on Yo’LLaVA and MyVLM datasets for evaluating the proposed framework and comparison with the existing state-of-the-art verifies their claims. In addition, elaborate ablation and analysis of robustness of the various parameters of the approach are also presented.